# Sequential metamaterials with alternating Poisson's ratios

Amin Farzaneh[1], Nikhil Pawar[1], Carlos M. Portela [2] & Jonathan B. Hopkins [1✉]

Mechanical metamaterials have been designed to achieve custom Poisson's ratios via the deformation of their microarchitecture. These designs, however, have yet to achieve the capability of exhibiting Poisson's ratios that alternate by design both temporally and spatially according to deformation. This capability would enable dynamic shape-morphing applications including smart materials that process mechanical information according to multiple time-ordered output signals without requiring active control or power. Herein, both periodic and graded metamaterials are introduced that leverage principles of differential stiffness and self-contact to passively achieve sequential deformations, which manifest as user-specified alternating Poisson's ratios. An analytical approach is provided with a complementary software tool that enables the design of such materials in two- and three-dimensions. This advance in design capability is due to the fact that the tool computes sequential deformations more than an order of magnitude faster than contemporary finite-element packages. Experiments on macro- and micro-scale designs validate their predicted alternating Poisson's ratios.

[1] Department of Mechanical and Aerospace Engineering, University of California, Los Angeles, Los Angeles, CA 90095, USA. [2] Department of Mechanical Engineering, Massachusetts Institute of Technology, Cambridge, MA 02139, USA. ✉email: hopkins@seas.ucla.edu

Mechanical metamaterials (i.e., architected materials)[1] can be engineered to achieve behaviors that predominantly stem from their architecture instead of their constituent material. Auxetic (i.e., negative Poisson's-ratio) behavior is one of the earliest[2,3] and most popular[4–10] behaviors engineered by mechanical-metamaterial designers since such behaviors are desirable for a variety of applications but are rarely achieved by natural monolithic materials. Unlike such traditional materials, which typically exhibit positive Poisson's ratios, auxetic materials expand laterally when they are stretched but contract when they are compressed.

Whereas some traditional materials do naturally exhibit negative[11–13] or even zero[14] Poisson's ratios, many two-dimensional (2D)[15–19] and three-dimensional (3D)[20–23] mechanical metamaterials have been engineered over the past four decades to achieve a broader range of both negative and positive Poisson's ratios that can be geometrically tailored to meet the demands of numerous applications. Medical applications include hip replacement implants that promote osseointegration by inducing strain in surrounding bone upon walking[24] and smart bandages with pores that passively open to release medication when swelling occurs[6]. Structural applications include indentation-resistant materials that contract around impacting objects[4,5], self-anchoring fasteners and screws[25], and materials that passively change their shape, size, or surface contour in response to particular loads[4–7]. Additionally, auxeticity has been employed to realize materials with highly tunable phononic bandgaps[26,27], transmission materials that improve actuator resolution and sensor sensitivity[28], and color-changing materials with colored voids that alter their size and thus visual prevalence as their lattices are loaded[29].

Despite the large number of engineered Poisson's-ratio metamaterial designs, which have been proposed in the last four decades, the field remains rich with new and creative innovations. Some of these innovations include the incorporation of buckling[30–32], origami[33–37], and kirigami[38,39] to achieve increasingly demanding Poisson's-ratio behaviors. Others have used the random placement of flexible elements[40], fibers[41,42], or pores[43] to achieve stochastically engineered negative or zero Poisson's ratio behaviors. Graded designs that consist of spatially varying Poisson's-ratio architectures have also been explored[44,45]. Some researchers have learned how to achieve desired Poisson's ratios that remain unchanged regardless of the direction in which the lattice is loaded[46,47] or how much the lattice is deformed[48–51]. Still others have incorporated multistability within auxetic metamaterials to study how wavefronts dynamically propagate within their lattices[52]. Some designs can even be programmed to achieve Poisson's ratios that can be changed from positive to negative or vice versa in response to a variety of external stimuli ranging from heat, ultraviolet light, magnetic fields, or moisture[53–55].

Although designs currently exist that achieve a gradually increasing or decreasing Poisson's ratio as a function of deformation magnitude[30,48,49], no designs can abruptly change the direction of their lateral deformations by alternating between positive and negative Poisson's ratios when loaded along a single unchanging direction. Moreover, no designs can achieve Poisson's ratios that spatially vary along the lattice geometry while also alternating in a desired time-ordered sequence. Such capabilities would enable compliant transmission mechanisms to transform a single-actuator input into multiple synchronized actuator outputs. These outputs could be tuned to achieve different but coupled displacement magnitudes and directions, which when cycled would manifest as different amplitudes and frequencies. Thus, such mechanisms could passively multiply an actuator's driving frequency (e.g., double it) while also improving its displacement resolution. Moreover, materials with spatially varying Poisson's ratios that also alternate in a carefully orchestrated time-ordered sequence would enable new means of single-actuator locomotion (i.e., one actuator could cause the material to undulate similar to the complex walking motion of a caterpillar). Such materials could also be used to assist mechanical logic devices[56] toward performing simple but critically reliable computations in potentially harsh environments where electronics are likely to fail. When positioned and loaded correctly, for instance, such a material could be used like a customized key in that it could trigger a series of flexure-based bi-stable logic-gate inputs[56] with the necessary deformation combination to perform the mechanical logic required to withdraw a bomb's fail-safe. In this way, the bomb wouldn't be capable of inadvertently exploding prior to being armed using the metamaterial key.

This work introduces a metamaterial that enables such applications via user-specified alternating Poisson's ratios. The concept is inspired by the inward facing beams (colored red) of the well-known negative-Poisson's-ratio honeycomb[3–7] shown in Fig. 1a,

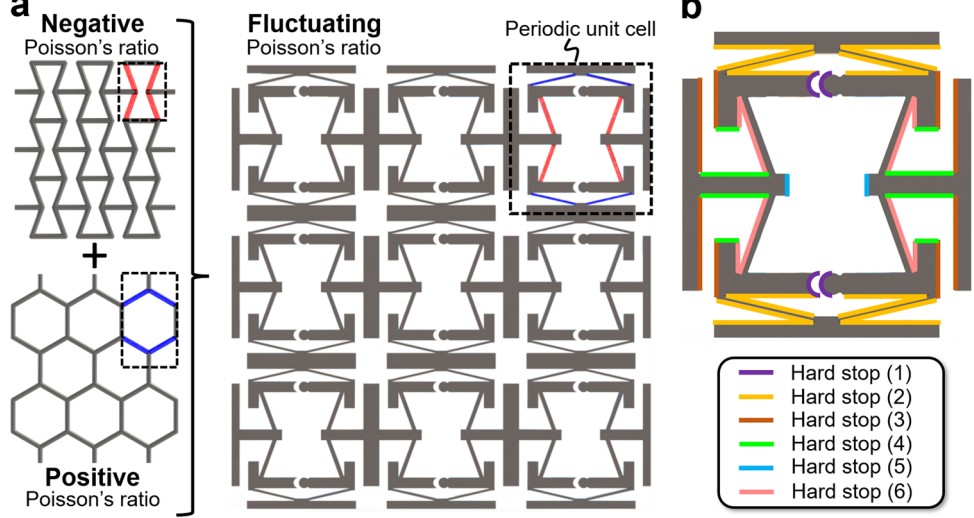

**Fig. 1 Sequential metamaterial that achieves an alternating Poisson's ratio. a** The concept is inspired by the inward (red) and outward (blue) facing beams of the famous negative and positive Poisson's-ratio honeycomb designs, respectively. **b** The design achieves its alternating Poisson's ratio due, in part, to six hard stops that collide and then redirect the motions of each unit cell rigid bodies.

and the outward facing beams (colored blue) of the equally popular positive-Poisson's-ratio regular honeycomb[57]. Although others have mixed the use of these two honeycomb cell designs for other purposes[58–60], here we combine different aspects of each honeycomb cell to produce a unique metamaterial design (Fig. 1a) with a Poisson's ratio that can alternate as a function of loading strain (note the similarly inward and outward facing beam pairs shown red and blue respectively in the design's periodic unit cell). The design is able to abruptly alternate its Poisson's ratio when subjected to a steady loading strain due primarily to collisions that occur between various combinations of the 6 carefully designed self-contacting hard stops labeled in Fig. 1b. By tuning the geometric parameters (e.g., thicknesses and lengths) that define these self-contacting bodies as well as the beam pairs that connect them, the bodies will displace and collide in a specific time-ordered sequence resulting in desired fluctuations in the bulk lattice's Poisson's ratio. Although the concept of time-ordered deformations achieved via self-contacting hard stops and different-stiffness flexible elements has previously been applied to enable metamaterials that sequentially fold[61,62], this work applies a similar concept but for the unique purpose of achieving metamaterials that exhibit alternating Poisson's ratios. Herein a MATLAB tool is provided to enable the design of such metamaterials so that they achieve the desired alternating Poisson's-ratio response to a given strain input load. The tool's theory is verified using finite element analysis (FEA) and validated using experimental measurements collected from macro-scale 2D unit cells. Principles are introduced and demonstrated for stacking different rows of identical unit cell designs in series to achieve Poisson's ratios that alternate both temporally and spatially along the resulting graded material's lattice. Finally, 3D versions of the design are introduced, fabricated, and tested at the micro-scale.

## Results

**Poisson ratios that alternate with strain.** FEA was used to computationally demonstrate that the metamaterial concept proposed in Fig. 1 could be tuned to exhibit a desired alternating Poisson's ratio. For the case of a lattice made of Teflon (i.e., a Young's modulus of 0.3657 GPa and a Poisson's ratio of 0.46) and with geometric parameters (defined in Fig. 2) set to those of the specific design shown in Fig. 3a (i.e., Design I with specific values provided in the Design I column of Supplementary Table 1), the lattice's response to a sinusoidal strain load is shown. Note from Fig. 3a that as the unit cell is loaded with a sinusoidal strain along the y-axis with a 15% strain amplitude (shown gray), the cell's side tabs will displace as the cell deforms with a resulting x-axis strain (shown green) that alternates with twice the frequency as the input load. The periodic lattice's Poisson's ratio can be calculated by dividing that x-axis strain with the negative of the y-axis loading strain. The resulting Poisson's ratio is plotted in Fig. 3a as the dashed purple line. Note from the plot that the FEA results were calculated for a single sinusoidal loading cycle with quasistatic loading conditions, and plotted over a period of 60 min to allow for comparison with other designs and experimental data.

The Poisson's ratio alternates primarily according to two factors—(i) different-stiffness beams and (ii) engineered hard stops. Note that the outward facing beams shown blue in Fig. 1a are more compliant than the inward facing beams shown red in the same figure. Thus, when the top tab is initially pulled upward, as long as the bottom tab is held fixed, the outward facing beams deform significantly more than the inward facing beams (since they experience the same loading force) resulting in the side tabs of the cell initially pulling inward. This occurs until the hard stops labeled (1) in Fig. 1b make contact with themselves at which point the side tabs immediately change direction and begin to move outward while the top tab continues to be pulled upward. Note that unlike the other hard stops in the design, Hard stop (1) is unique in that both pairs consist of a mating circular feature that fits inside of an extruded half cylinder, which causes both contacting bodies to rotate relative to one another about an axis located at the center of the circular feature. Thus, when engaged, Hard stop (1) generates friction, which can lead to wear, hysteresis, and loss of repeatability. As such, designs that require the engagement of Hard stop (1) to function, should be fabricated using a material with a low coefficient of friction (e.g., Teflon) to minimize these effects.

In addition to exhibiting an alternating x-axis strain when the unit cell is loaded in tension along the y-axis, the cell also exhibits an alternating but oppositely directed x-axis strain when it is loaded in compression along the same axis (Fig. 3a). When the top tab is initially pushed downward, the outward facing beams

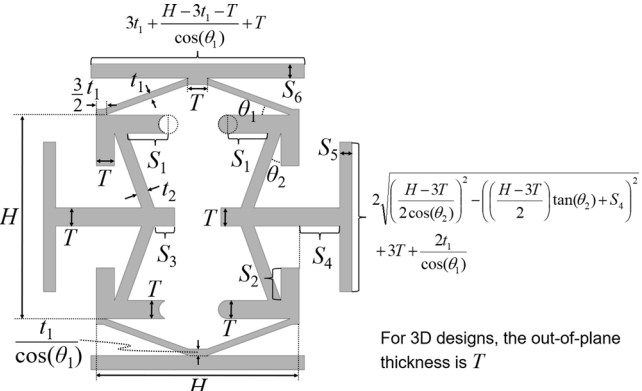

$$3t_1 + \frac{H - 3t_1 - T}{\cos(\theta_1)} + T$$

$$2\sqrt{\left(\frac{H-3T}{2\cos(\theta_2)}\right)^2 - \left(\left(\frac{H-3T}{2}\right)\tan(\theta_2) + S_4\right)^2} + 3T + \frac{2t_1}{\cos(\theta_1)}$$

For 3D designs, the out-of-plane thickness is $T$

**Fig. 2 Geometric parameters that define the proposed metamaterial unit cell concept.** The parameters can be tuned to achieve different alternating Poisson's ratios over time according to how the cell is loaded. Note that the design is symmetric about the dotted red horizontal line and is almost symmetric about the dotted blue vertical line except for the circular hard stops labeled Hard stop (1) in Fig. 1b.

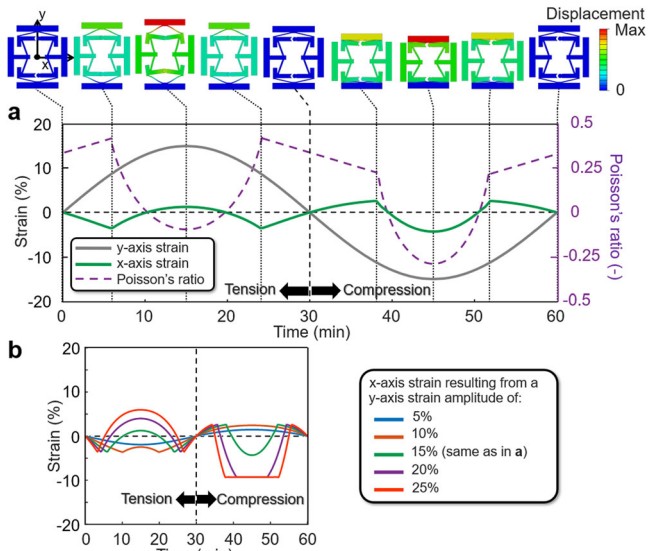

**Fig. 3 Finite-element-analysis (FEA) generated plots that demonstrate Design I's sequential deformations given Teflon material properties.**
**a** Plot showing how 9 still frame images of the Design I cell correspond with the cell's sequential deformations for a sinusoidal y-axis input strain with an amplitude of 15%. **b** Plot showing the cell's x-axis strain responses to five different y-axis sinusoidal loading amplitudes ranging from 5% to 25%.

again deform significantly more than the inward facing beams since they are more compliant and thus the side tabs of the cell initially push outward. This occurs until the hard stops labeled Hard stop (2) in Fig. 1b make contact with themselves at which point the side tabs immediately change direction and begin to move inward as the inward facing beams exclusively deform while the top tab continues to be pushed downward. Note from Fig. 2 that the tab lengths are geometrically constrained such that they will always appropriately behave as hard stops, when necessary, without ever interfering or colliding with each other as cell designs are deformed. A sequence of 9 still frame images showing the cell being sinusoidally deformed in both tension and compression is provided in Fig. 3a where each corresponding frame is shown labeled in the plot. An animated video of the FEA simulation is also provided in Supplementary Movie 1.

Equally interesting to the fact that the proposed design can be tuned to achieve desired alternating Poisson's ratios, is the fact that the design's $x$-axis strain response is dependent on the magnitude of the $y$-axis loading strain. The plot shown in Fig. 3b demonstrates this dependence for five sinusoidal $y$-axis loading strains with amplitudes ranging from 5% to 25% strain. Animated videos showing the FEA simulations of all five loading scenarios are provided in Supplementary Movie 1. Note that when the cell is sinusoidally loaded with only 5% strain amplitude, the resulting $x$-axis strain does not alter its frequency of oscillation but rather moves in the opposite direction of the input load with a smaller amplitude. When the cell is sinusoidally loaded with a strain amplitude >20%, however, the resulting $x$-axis strain not only alternates back and forth with a different frequency than the input, but multiple hard stops are sequentially engaged during compression (i.e., Hard stop (2) and then Hard stop (3) labeled in Fig. 1b), which results in an $x$-axis strain that eventually remains constant while the top tab is being pushed downward to its maximum displacement.

By changing the geometric parameters defined in Fig. 2 other versions of the design of Fig. 1 can be tuned to achieve different alternating Poisson's-ratio behavior. If, for instance, the beam stiffness values of Design I are modified so that the outward facing beams shown blue in Fig. 1a are made stiffer than the inward facing beams shown red in the same figure, a new design (i.e., Design II) can be created that achieves an oppositely alternating $x$-axis strain response (Fig. 4). Specific values for the geometric parameters of Design II are provided in the Design II column of Supplementary Table 1. Note that the $x$-axis strain response of Design II (shown purple in Fig. 4) to a sinusoidal $y$-axis strain load with an amplitude of 15% (shown gray) alternates in the opposite directions as the $x$-axis strain response of Design I (shown green) for the same loading conditions.

The lattice design introduced in this work can also exhibit alternating Poisson's-ratio behavior when loaded in different directions. Note from Fig. 4 that Design III achieves an alternating $x$-axis strain response (shown brown) to the $y$-axis

strain input load (shown gray) even when the cell is rotated 90º on its side. Specific values for the geometric parameters of Design III are provided in the Design III column of Supplementary Table 1. Although different versions of the design concept of this work can be made to achieve alternating Poisson's-ratio behaviors when loaded in multiple directions, such versions will not be isotropic (i.e., the Poisson's-ratio behaviors will not alternate the same when the design is loaded in different axes). Animated FEA videos showing Designs II and III being loaded according to the scenario plotted in Fig. 4 are provided in Supplementary Movie 2.

**Analytical MATLAB tool.** A MATLAB tool was created to efficiently calculate how the Poisson's ratio of different versions of this work's metamaterial design alternates in response to a sinusoidal quasistatic load. The tool's graphical user interface (GUI) is shown in Supplementary Fig. 1. When the tool is launched (Supplementary Fig. 1a), it prompts users to enter (i) the properties of the lattice's desired constituent material (i.e., its Young's modulus and Poisson's ratio), (ii) the geometric parameters that define the lattice's repeating unit cell, (iii) the amplitude of the input strain load that will sinusoidally drive the lattice, and (iv) a resolution number, called the accuracy number, that when made larger increases the accuracy and smoothness of the resulting plots, but also increases the computational time required to generate the plots. After these parameters are entered, the tool then displays a plot (Supplementary Fig. 1b) of the resulting design's $x$-axis strain response (shown blue) along with the design's corresponding Poisson's ratio (shown dashed purple). It also shows an animation of the unit cell deforming with a vertical black line sweeping over the plot corresponding to each frame of the deformed cell. The tool then allows users to either re-enter new parameters to generate a different cell design or save the animation of the existing cell design as a .gif file. The MATLAB tool is available for download (see Supplementary Software 1) and a video showing the animated version of Supplementary Fig. 1b is provided in Supplementary Movie 3. Instructions for running the MATLAB tool are also available in Supplementary Information.

The MATLAB tool enables the design of metamaterials with Poisson's ratios that alternate as desired due to the efficiency in which the tool is able to analytically calculate the large-deformation response of such materials to a sinusoidal strain load. Other FEA-based approaches are too computationally expensive and time-consuming to calculate the large-deformation Poisson's-ratio behavior of this work's metamaterial concept fast enough to enable their design. Thus, as the following section demonstrates, the analytical tool introduced in this paper is currently the only practical option for analyzing the proposed metamaterials of this work fast enough to enable their design. A description of the MATLAB tool's underlying theory along with its assumptions is provided in the "Methods" section.

**FEA verification and experimental validation.** Despite its simplifying assumptions, the MATLAB tool analytically predicts the design's large-deformation behavior with impressive accuracy. Figure 5a shows a plot that compares the results of the MATLAB tool (shown blue) with two large-deformation FEA simulations (shown green and orange) and with experimental data (shown red) measured by loading a Design I cell made of Teflon in an Instron mechanical testing machine. The MATLAB and FEA results were calculated using the properties of Teflon provided previously. The smallest accuracy number that still achieved smooth plots (i.e., 60) was applied to the MATLAB tool and Abaqus was used to perform the FEA simultaions. Hexahedral elements were applied to the unit cells of the FEA simulation with

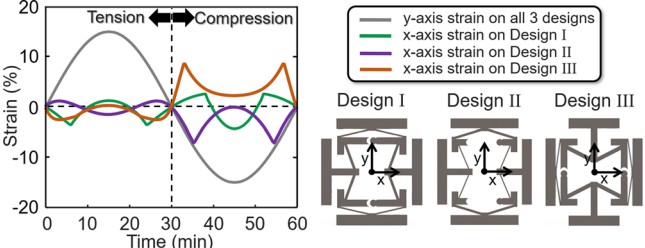

**Fig. 4** The $x$-axis strain response of three different cell designs (i.e., Design I, II, and III) made of Teflon to a sinusoidal $y$-axis strain load with an amplitude of 15%.

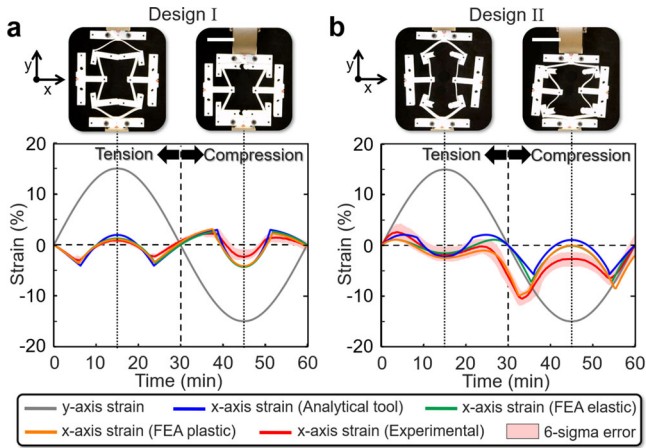

**Fig. 5** Comparisons between the analytical MATLAB tool, elastic and plastic finite element analysis (FEA) simulations, and experimental test results for **a** Design I and **b** Design II unit cells. Scale bars in **a** and **b**, 5 cm.

an out-of-plane thickness of 10 mm. The FEA simulation, shown in green, was calculated assuming that the cell's material properties are linear elastic, that no gravity is present, and that the cell is embedded within an infinitely large lattice so that its two side tabs are constrained to remain vertically oriented as they displace (all of the FEA simulations provided in Figs. 3 and 4 were generated with these same assumptions). Note from Fig. 5a that this FEA elastic simulation is almost identical to the analytical results predicted by the MATLAB tool. The other FEA simulation, shown orange, varies more significantly from the results of the tool since that simulation assumes that the material properties are not linear elastic (i.e., it assumes the Teflon stress–strain curve provided in Supplementary Fig. 2), that gravity is pulling downward along the y-axis direction shown, and that the cell is not a part of a lattice so that its side tabs are allowed to rotate. These assumptions most closely mimic the experience of the fabricated single-cell version, which is shown being tested in the Instron machine in two states of deformation corresponding with the plot of Fig. 5a.

A similar comparison between the analytical MATLAB tool of this paper, the elastic and plastic FEA simulations, and the experimental measurements applied to Design II was also performed as shown in Fig. 5b. Note that although the MATLAB tool (shown blue) and FEA elastic simulation (shown green) results are similar and the FEA plastic simulation (shown orange) and experimental (shown red) results are also similar, both pairs of similar results diverge from each other after the cell is tensioned to its maximum strain. The reason for this discrepancy is that the 15% strain load was sufficiently large to cause significant yielding within the Design II cell, which was not accounted for in the analytical tool or the FEA elastic simulation results. Note that yielding is also primarily responsible for the asymmetry observed in both the tension and compression regions of the FEA plastic and experimental plots. Discrepancies between the FEA plastic and experimental results shown in the compression region of Fig. 5b are primarily due to fabrication and fixturing imperfections, which led to asymmetric buckling when Design II's cell was compressed (see the top right image in Fig. 5b). Thus, as long as minimal or no yielding occurs in the cell designs of this work and as long as sufficient care is taken to fabricate and fixture the cells correctly, the analytical MATLAB tool can accurately predict their alternating Poisson's ratios. Animations of the FEA plastic simulations of both Designs I and II alongside corresponding videos of the fabricated cells being tested are provided in Supplementary Movie 4. Details pertaining

to how the cells shown in Fig. 5 were fabricated and tested are provided in the "Methods" section.

To compare the computational efficiency of the MATLAB tool with FEA, the time required to generate the analytical and FEA plots of Fig. 5 were measured. Using a standard desktop computer, the MATLAB tool generated the blue plots of Fig. 5a and b in 21 and 15.6 s, respectively. Using the same computer, Abaqus generated the green FEA elastic plots of Fig. 5a, b in 1670 and 619 s, respectively. Abaqus also generated the orange FEA plastic plots of Fig. 5a, b in 2002 and 720 s, respectively. Note that the MATLAB tool ranged from 39.7 to 95.3 times faster than the FEA simulations, which is significantly more than an order of magnitude more computationally efficient.

Data was also collected and compared against the FEA plastic simulations of Designs I and II when they were both rotated 90° on their sides as shown in Supplementary Fig. 3. These experiments were conducted to demonstrate that the same cell's response to a particular loading scenario differs substantially depending on the direction in which the cell is loaded. Animations of these FEA plastic simulations alongside corresponding videos of the fabricated cells being tested are also provided in Supplementary Movie 5.

**Graded lattices**. In addition to being able to alternate its Poisson's ratio temporally, the metamaterial concept of this work can also be made to alternate its Poisson's ratio spatially along its geometry as long as the lattice is graded with different (i.e., aperiodic) rows of repeating cell designs. Consider, for instance, the graded metamaterial design shown in Fig. 6. Its lattice consists of four different cell designs shown in Fig. 6a, which are each repeated four times within four different rows as shown in Fig. 6b. The geometric parameters of each of the four cell designs defined in Fig. 2 are provided in Supplementary Table 2. Each of these cells were designed using the analytical MATLAB tool to independently achieve the x-axis strain response shown in Fig. 6c to a half-cycle sinusoidal strain load with a tensile amplitude of 13% (shown gray). The plot of Fig. 6c accurately reflects each unit cell's response as long as they are separately loaded individually or belong within a periodic lattice that consists entirely of the same repeating cell design. Note, however, from Fig. 6d that when the cell designs are joined together within the graded metamaterial lattice of Fig. 6b, the resulting material responds to the same load with a similar but notably altered behavior. The plot of Fig. 6d is the measured data collected from the lattice of Fig. 6b being loaded in an Instron testing machine. Although vertical flexures (Fig. 6b) were added to the upper and lower tabs of the top and bottom rows of the lattice so that the lattice's Poisson's-ratio behavior would be minimally affected by the loading constraints of the Instron, the results of Fig. 6d differ from the intended response of Fig. 6c for two primary reasons. First, each row of cell designs frustrates the different expansion and contraction attempts of its neighboring rows because the rows are directly connected together along their upper and lower tabs. Second, each of the four cell designs of Fig. 6a exhibits a different overall stiffness and thus, when the graded metamaterial is loaded with a desired strain, each row experiences a different amount of strain (i.e., the compliant rows deform more than the stiff rows). Thus, no row within the graded metamaterial of Fig. 6b experiences the 13% strain amplitude that was assumed for each individual cell when they were being designed using the MATLAB tool to produce the plot of Fig. 6c.

Therefore, employing the MATLAB tool to accurately design graded metamaterials that achieve desired temporally and spatially alternating Poisson's-ratio behaviors requires consideration of the following principles. First, designers could separate

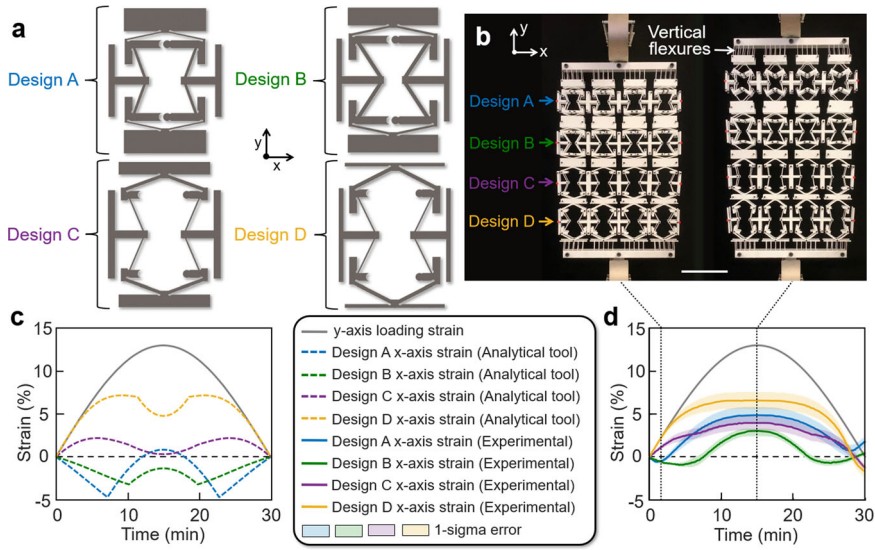

**Fig. 6 Graded metamaterials consisting of rows of different unit cell designs can be made to achieve both temporally and spatially alternating Poisson's ratios. a** Four different cell designs were used within each row of (**b**) a graded metamaterial design example, which was fabricated from Teflon using a laser cutter. **c** The *x*-axis strain response of each independent unit cell design in (**a**) for a given strain load as predicted using the analytical MATLAB tool when the cells are not joined together. **d** The experimentally measured strain response of each row of cell designs to the same load when joined together within the metamaterial lattice of (**b**). Scale bar in (**b**), 10 cm.

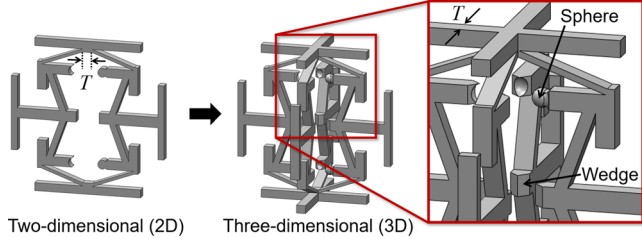

**Fig. 7 Two-dimensional (2D) designs can be transformed into three-dimensional (3D) versions that achieve the same alternating Poisson's ratio.** These 3D versions are generated by rotating a copy of their 2D version 90° about its central axis. Some 2D-version hard stops need to be adapted for the 3D version to function with the same alternating Poisson's ratio.

each row of different cell designs using vertical flexures (similar to those labeled in Fig. 6b) between their upper and lower tabs to mitigate the issue of their different row expansions and contractions frustrating one another's desired deformations. Second, designers could either make sure that the cells that constitute each row exhibit the same stiffness as their *x*-axis strain responses are being tuned using the MATLAB tool, or they could account for the portion of the overall lattice loading strain that each row of cells will experience if their stiffnesses are different. Note that each row of cells will always experience the same loading force since they are serially stacked.

Details pertaining to how the lattice shown in Fig. 6b were fabricated and tested are provided in the "Methods" section. An animation of an FEA simulation of the lattice (assuming gravity and the nonlinear plastic properties of Teflon provided in Supplementary Fig. 2) is shown in Supplementary Movie 6, alongside a corresponding video of the fabricated lattice being loaded by the Instron testing machine.

**Three-dimensional version.** This work's proposed concept for achieving alternating Poisson's ratios is not limited to 2D extrusions only. If a 2D cell design is copied and rotated 90° about its central axis as shown in Fig. 7, a 3D unit cell version is

generated. The 2D design's extrusion thickness must, however, be equal to the *T* parameter labeled in Fig. 2 for the resulting 3D design to be geometrically compatible. The 3D version will possess the same large-deformation Poisson's-ratio behavior as its 2D analog, but it will be twice as stiff. Thus, the MATLAB tool provided in this paper enables the design of both 2D and 3D metamaterials that achieve desired alternating Poisson's ratios. For the 3D designs to function the same as their 2D analogs, however, it's necessary that the geometry of some of the 2D hard stops be altered. The circular feature labeled Hard stop (1) in Fig. 1b should, for instance, be changed from a cylinder to a sphere and the three other mating surfaces should be triangular wedges with inverse sphere shapes cut from their ends as shown in Fig. 7. Additionally, the flat features labeled Hard stop (5) in Fig. 1b should be changed to triangular wedges as shown in Fig. 7.

**Micro-scale validation.** A 3D micro-scale version of the proposed concept was additively fabricated from Ip-Dip photoresist (Nanoscribe GmbH) as a single unit cell using a two-photon lithography system (Nanoscribe PPGT). The 3D cell, shown in Fig. 8a and Supplementary Fig. 4 was fabricated with a custom grip on its top surface to enable the cell to be mechanically tested in both tension and compression. The geometric parameters of the fabricated cell as defined in Supplementary Fig. 5 are provided in Supplementary Table 3. Note that some hard stops, many of which are irrelevant to the functionality of the particular design, were eliminated or simplified to better facilitate the cell's fabrication. Some features, which are unfortunately important to the cell's performance, had to be tapered for similar fabrication purposes. An in situ nanoindenter (inSEM II, Nanomechanics) was used to stretch and compress the cell along the *y*-axis over two complete cycles in a period of 2000 s as shown by the gray line in the plot of Fig. 8b. The resulting *x*-axis strain response, which was determined by tracking the displacements of the cell's side tabs using image recognition software, is shown red in Fig. 8b. Note that although the cell's *x*-axis strain response successfully alternates as intended, a small degree of yielding and thus hysteresis occurs in the cell as indicated by a slightly differing second deformation cycle compared to the first. A video

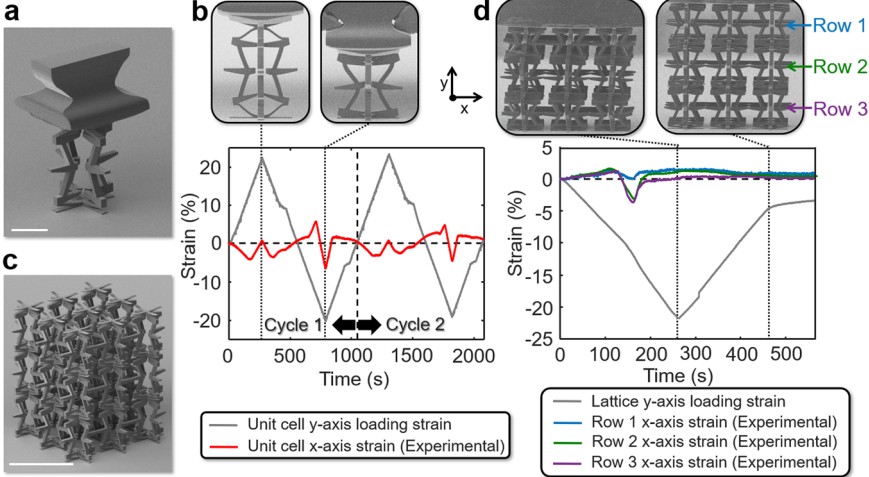

**Fig. 8 Fabrication and experimental validation of a micro-scale three-dimensional (3D) unit cell and lattice that achieves an alternating Poisson's ratio.**
**a** An additively fabricated micro-scale unit cell with a custom grip to enable its loading in both tension and compression. **b** The alternating *x*-axis strain response of the cell being cyclically loaded in tension and compression along the *y*-axis. **c** An additively fabricated micro-scale 3 × 3 × 3 lattice of the repeating unit cell design. **d** The alternating *x*-axis strain response of each row of cells within the lattice to a compressive *y*-axis strain load. Scale bar in (**a**), 50 μm, and in (**c**), 200 μm.

showing the cell being loaded next to a corresponding strain-versus-time plot is provided in Supplementary Movie 7.

A 3 × 3 × 3 micro-scale lattice of the same cell design was also additively fabricated from the same material using the same system as shown in Fig. 8c. Since no custom grip was able to be printed on its top surface without forcing all the cells within the lattice's top layer to be joined together, the lattice was only able to be tested in compression along the *y*-axis. The *x*-axis strain response of each row of cells is labeled in the plot of Fig. 8d. Each row has a different *x*-axis strain response primarily because the friction imposed by the compression tip on the lattice's top and bottom rows caused each row to frustrate each other's natural deformations. This frustration effect in conjunction with the fact that (i) the top and bottom tab hard stops between each layer curled in different and undesired ways during fabrication, and (ii) each cell was not identically printed due to fabrication imprecision, caused the top and bottom layers of the lattice to misalign and buckle. This undesired and asymmetric buckling caused the expected *x*-axis strain fluctuation to terminate prematurely, which is why the blunted dip in the lattice's *x*-axis compressive strain response, shown in Fig. 8d, appears to occur earlier than the corresponding dip produced by the single unit cell shown in Fig. 8b. Thus, with improved fabrication capability and for lattices that consist of many more unit cells than the small 3 × 3 × 3 lattice, the 3D metamaterial designs proposed in this paper are expected to more closely approach the designed behaviors of their single repeating cells. A video showing the 3 × 3 × 3 lattice of repeating cells being loaded next to a corresponding strain-versus-time plot is provided in Supplementary Movie 8. Other larger (e.g., 5 × 5 × 5 cell) lattices that consist of a different but repeating unit cell design were also fabricated as shown in Supplementary Fig. 6.

## Discussion

A metamaterial concept was introduced that uses the principles of differential stiffness and self-contacting hard stops to achieve Poisson's ratios that alternate as desired from positive to negative values or vice versa as the material is strained along multiple axes in both tension and compression. A MATLAB tool was created to enable the design of both 2D and 3D versions of the concept due to the fact that the tool's custom-developed analytical theory is

more than an order of magnitude faster than other FEA-based computational approaches. The tool is only accurate if the repeating unit cells that constitute the metamaterial do not appreciably yield as they are loaded. The tool was verified using FEA and validated experimentally using multiple macro-scale designs, which were fabricated using assembled sheets of laser-cut Teflon. A graded lattice consisting of rows of different unit cell designs was also fabricated and tested to validate the concept that a material's Poisson's ratio can be made to alternate both temporally and spatially along its loading direction to transform a single actuated signal into multiple output signals with coupled but different amplitudes and frequencies. Finally, 3D lattices consisting of micro-scale unit cells were additively fabricated and tested to demonstrate that the concept proposed can be extended to achieve alternating Poisson's ratios within practical 3D volumes and with architected features that are scaled to their intended size. The bulk shape of such lattices, the precision (i.e., repeatability) of the approach used to fabricate such lattices, and the boundary and loading conditions imparted on such lattices can all have a significant effect on the overall Poisson's-ratio behavior and must be considered in the design processes.

## Methods

**Theory and assumptions underlying the MATLAB tool**. The MATLAB tool is especially efficient due to several simplifying assumptions. It models the design's cell as a collection of compliant rectangular-prism-shaped beams (shown gray in Supplementary Fig. 7a) that join rigid-body nodes (shown black in Supplementary Fig. 7a) together. A twist-wrench stiffness matrix is constructed to represent the elastomechanic properties of the cell using Bernoulli–Euler beam elements according to the theory provided in Hopkins et al. [63]. The cell's upper tab is loaded with an infinitesimal force and the twist-wrench stiffness matrix is used to calculate where all the rigid bodies incrementally move in response with its lower tab held fixed. Once their positions are updated, the beams are reconstructed to join the newly positioned rigid bodies together as if the beams had never been deformed. This process repeats increment after increment until the top tab moves the amount prescribed by the strain amplitude in both tension and compression directions. If any of the rigid bodies touch at hard stops in the process, the bodies are effectively fused together such that they are labeled with the same number as shown by the bodies labeled *B7* and *B14* in Supplementary Fig. 7b. All the bodies are re-labeled and the twist-wrench stiffness matrix is updated accordingly. The details of the theory are provided in the script of the MATLAB code provided in Supplementary Software 1.

**Fabrication and testing details for the cells in Fig. 5**. The unit cell designs of Fig. 5 were fabricated by laser cutting two copies of each cell design from 1/8-inch

sheets of Teflon as shown in Supplementary Fig. 8a. Separator posts were used to join the resulting pieces together so that the final unit cells would not buckle out-of-plane when they are loaded in compression. Red-dot sticker sensing markers were placed on the middle far edge of each design's tabs as shown in Supplementary Fig. 8b so that image recognition software could be used to track their locations as the Instron testing machine loaded the gripping tabs with the prescribed strains (Supplementary Fig. 8a, c). The six-sigma red shaded error regions in the plots of Fig. 5 were calculated from the standard deviations of five full loading cycles performed on each cell design.

**Fabrication and testing details for the lattice in Fig. 6b**. The lattice of Fig. 6b was also assembled using two laser-cut sheets of 1/8-inch Teflon (Supplementary Fig. 9). The sheets were separated by longer posts than those used for the individually fabricated unit cells to prevent the larger lattice from buckling out of plane. Red-dot sticker sensing markers were placed on the center edge of each row's outermost side tabs to enable image recognition software to measure the response of each row's deformations to the strain load imposed by the Instron testing machine.

## Data availability

The authors declare that the data supporting the findings of this study are included in the main text, the Supplementary Information, the GitHub repository link above, or are available from the corresponding author upon request.

## Code availability

Supplementary Software 1 is available using a GitHub repository link provided at the following URL: https://github.com/aminfno/Metamaterial. Authors: A.F. and J.B.H., Title: Sequential Metamaterials with Alternating Poisson's Ratios, Repository Name: aminfno/Metamaterial, Year: 2021. To launch the tool, download all five MATLAB files and locate them in the same directory on your computer. Then open the directory and click on the Metamaterial.m file and run it in MATLAB. The tool's graphical user interface (GUI) will then be launched.

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

## Acknowledgements
This work was supported by AFOSR under award number FA9550-18-1-0459 (J.B.H.). The authors acknowledge program officer Byung "Les" Lee. Prof. Julia R. Greer is also thanked for allowing the use of laboratory equipment at Caltech to fabricate and test the micro-scale designs.

## Author contributions
A.F. wrote the GUI script for the MATLAB tool, performed the paper's FEA, and processed the macro-scale mechanical testing data. N.P. laser cut, assembled, and helped perform the macro-scale mechanical testing. C.M.P. fabricated, imaged, and mechanically tested the micro-scale unit cell and its lattice. J.B.H. conceived the paper's concept, generated the paper's designs, produced the analytical theory for the analysis of the paper's large-deformation designs, coded the theory into the MATLAB tool, made the paper's figures, wrote the paper, and managed the project.

## Competing interests
The authors declare no competing interests.
