## [Peer Review File · Nature Communications]

Title: Sequential Metamaterials with Alternating Poisson's RatiosREVIEWER COMMENTS

Reviewer #1 (Remarks to the Author):

Review Report

The authors conducted an interesting work by using analytical, numerical and experimental methods. The designed metamaterials could exhibit tunable Poisson's ratio under tension and compression. The proposed metamaterials may have the potentials for dynamic shape-morphing application. The manuscript is well organized and the content is clearly explained.

But, considering the content of this work is actually related to the topic of "Auxetic Metamaterials with Tunable Mechanical Properties", the following references should be included and discussed:

1. Tunable compressive properties of a novel auxetic tubular material with low stress level (DOI: 10.1016/j.tws.2021.107882)
2. Tunable Auxetic Mechanical Metamaterials with "Arch-Shaped" Units ((DOI: 10.1002/pssr.201800376)
3. Experiments and parametric studies on 3D metallic auxetic metamaterials with tuneable mechanical properties (DOI: 10.1088/0964-1726/24/9/095016)
4. Smart metamaterials with tunable auxetic and other properties (DOI: 10.1088/0964-1726/22/8/084016)

The manuscript could be accepted after considering my comments.

Reviewer #2 (Remarks to the Author):

This paper presents a novel metamaterial design that can allow for fluctuating Poisson's ratios (from positive to negative and back, or vice-versa). The work presents the design with different parameters and shows how these affect the change in Poisson's ratios. An analytical MATLAB tool was also developed and is shown in the work. The tool uses screw and wrench theory to determine the deformations of the metamaterial design under a sinusoidal load. This tool, FEA (both elastic and elastic-plastic), and experimental data are compared to show the agreement. These metamaterials designs can also be connected sequentially to produce a mechanism that can potentially send multiple shifted signals from the same mechanism. Overall, the ideas presented in this work are novel. This work stands to be of great benefit to the area of engineering and materials. The paper is well-written and nicely organized for the most part. While overall this work is very good, there are a few things that should be addressed before publication. They are as follows:

- The time difference between the analytical tool and the FEA is discussed (being > 37 times faster). However, from inspection of Figure 4, it appears that there is some difference (not just time) in the accuracy of each model. A comparison of time difference and differences in error for the tool, FEA, and experiment would better help the reader to see why the presented analytic tool is as beneficial as

stated.

- In Figure 4, the FEA and analytical tool seem to compare well to the “Tension” portion of the experimental data, however, in both a and b in the “Compression” portion (near 45 min), none of the methods seem to compare well. What is occurring at this point that may be affecting this data so that the simulations aren’t accurately capturing the experimental phenomena? A discussion of this phenomena and why it is happening would help to strengthen the paper.
- On page 13, the reason for divergent results in the Design II plot (b) is discussed. The reason is that yielding occurred. Was an attempt to run the compression test separate from the tension test, to avoid this yielding, considered?
- Supplementary Figure 1 is referenced multiple times within the main body of the paper, which almost certainly requires the reader to review the supplementary material. With a figure being so heavily used in the main body of the paper, the paper may be better organized with Supplementary Figure 1 as a figure in the main body of the paper.

Reviewer #3 (Remarks to the Author):

The purpose of this manuscript may be summarized by a statement given in the introduction: “This work introduces a new kind of metamaterial that enables such applications via user specified fluctuating Poisson’s ratios.” Toward that the authors provide a very rigorous modeling effort, combined with manufacture and experimental validation. The model and implementation are of top quality and very comprehensive. The verification with FEA is appreciated, specifically how it is used to determine the working limits of the model. The 2D and 3D experimental effort shows that this design functions well across length scales, and in multiple spatial dimension. Overall I believe there is a contribution to the literature here, the topic is of interest to readers from a variety of fields, and accordingly I recommend only minor revisions, and some specific comments are listed below.

1) For some reason I think the authors can get a lot more traction by substituting the word fluctuating with something else. The definition I found of fluctuate is “rise and fall irregularly in number or amount.” The irregular part is what gives me second thoughts. I concede this isn’t necessarily how everyone will interpret the word, but this is just my opinion and does not need to be considered to heavily. Just think a different choice of word may be better, but I also cannot offer any appropriate suggestions. (The best I could think of was programmed mutation.)

2) Around line 121, the authors mention a 60 minute period to keep things quasi-static. Generally speaking, the default in a FEA is a static analysis, which does not include any inertial terms, and therefore does not include any dynamics. But to point it out might imply that a dynamic analysis was used, and I think that should be mentioned in the manuscript.

3) Around line 135 there is mention of hard stop 1 the only place where friction is expected. Is this friction an enemy to the desired behavior, or does it not matter and the choice of Teflon as the material not important?

4) Around line 202, the authors mention that the properties of the constituent material are needed for the matlab tool. I would prefer if the properties required were explicitly mentioned, for example is it the thermal conductivity that is needed? Or just the elastic modulus? It is generally obvious, but never explicitly written.

5) There is mention of a 37 times faster calculation when compared to standard FEA. But I also wasn't able to determine if that was an elastic-plastic simulation or just an elastic simulation. Or a static or dynamic analysis. Those FEA details would help to make the comparison a bit more measurable to readers.

6) Line 350 mentions an "in-situ panoindenter." Since this is on the cutting edge I actually did a double take before realizing its a simple typo.

RESPONSE TO REVIEWER COMMENTS

We are grateful to the reviewers for their insightful comments and for the time invested to help improve our paper. Reviewer comments are highlighted in dark blue and our responses follow. Page numbers are highlighted with the color used to highlight the corresponding revisions made on the corresponding page in the manuscript.

Reviewer #1

The authors conducted an interesting work by using analytical, numerical and experimental methods. The designed metamaterials could exhibit tunable Poisson's ratio under tension and compression. The proposed metamaterials may have the potentials for dynamic shape-morphing application. The manuscript is well organized and the content is clearly explained.

But, considering the content of this work is actually related to the topic of "Auxetic Metamaterials with Tunable Mechanical Properties", the following references should be included and discussed:

1. Tunable compressive properties of a novel auxetic tubular material with low stress level (DOI: 10.1016/j.tws.2021.107882)
2. Tunable Auxetic Mechanical Metamaterials with "Arch-Shaped" Units ((DOI: 10.1002/pssr.201800376)
3. Experiments and parametric studies on 3D metallic auxetic metamaterials with tuneable mechanical properties (DOI: 10.1088/0964-1726/24/9/095016)
4. Smart metamaterials with tunable auxetic and other properties (DOI: 10.1088/0964-1726/22/8/084016)

The manuscript could be accepted after considering my comments.

We thank the reviewer for the positive comments and for the additional references. We have revised the paper to include the references provided above (pp. 2,3,25,29).

Reviewer #2

This paper presents a novel metamaterial design that can allow for fluctuating Poisson's ratios (from positive to negative and back, or vice-versa). The work presents the design with different parameters and shows how these affect the change in Poisson's ratios. An analytical MATLAB tool was also developed and is shown in the work. The tool uses screw and wrench theory to determine the deformations of the metamaterial design under a sinusoidal load. This tool, FEA (both elastic and elastic-plastic), and experimental data are compared to show the agreement. These metamaterials designs can also be connected sequentially to produce a mechanism that can potentially send multiple shifted signals from the same mechanism. Overall, the ideas presented in this work are novel. This work stands to be of great benefit to the area of engineering and materials. The paper is well-written and nicely organized for the most part. While overall this work is very good, there are a few things that should be addressed before publication. They are as follows:

We thank the reviewer for the careful review, constructive comments, and generous assessment. We have revised the paper to address the discussed points below.

- The time difference between the analytical tool and the FEA is discussed (being > 37 times faster). However, from inspection of Figure 4, it appears that there is some difference (not just time) in the accuracy of each model. A comparison of time difference and differences in error for the tool, FEA, and experiment would better help the reader to see why the presented analytic tool is as beneficial as stated.

We have updated the manuscript to include a more thorough discussion of the time differences and the differences in error between the tool, FEA, and experimental results (pp. 11-14,14).

• In Figure 4, the FEA and analytical tool seem to compare well to the “Tension” portion of the experimental data, however, in both a and b in the “Compression” portion (near 45 min), none of the methods seem to compare well. What is occurring at this point that may be affecting this data so that the simulations aren’t accurately capturing the experimental phenomena? A discussion of this phenomena and why it is happening would help to strengthen the paper.

We added more discussion about the discrepancies between all the plots in Fig. 4 (pp. 12-14,14) and specifically we added an explanation of the reason for the divergence of the experimental data and the rest of the plots at the middle of the compression region (near 45 min) (pp. 14). Essentially the reason can be seen in the figure below. Due to fabrication and fixturing imperfections, the cells buckle asymmetrically in compression. This effect is not considered in the MATLAB tool or in FEA since they remain symmetric over the full loading cycle.

Design II Response to 15% y-axis strain amplitude

• On page 13, the reason for divergent results in the Design II plot (b) is discussed. The reason is that yielding occurred. Was an attempt to run the compression test separate from the tension test, to avoid this yielding, considered?

Unfortunately, the number of fabricated samples was limited so we were unable to perform the proposed test. However, due to the fact that an FEA simulation was generated for purely elastic properties and for plastic properties, we were able to isolated the theoretical effect of yielding over the full loading cycle. If the proposed compression test only had been performed on a freshly fabricated cell, we suspect that the results would match better but not entirely. Moreover, the result would still not be very symmetric since the cell also yields significantly during compression.

• Supplementary Figure 1 is referenced multiple times within the main body of the paper, which almost certainly requires the reader to review the supplementary material. With a figure being so heavily used in the main body of the paper, the paper may be better organized with Supplementary Figure 1 as a figure in the main body of the paper.

This is a good point which we agree would make things more convenient for the reader. However, given the part in the paper where the figure needs to first be referenced, the figure would either need to be part of the first concept figure or it would need to be a stand-alone figure by itself. We don’t want such a detailed distracting image being part of the initial concept figure and we have

already reached the allowable limit of figures in the paper to have it stand on its own. Nor do we feel like the figure is substantive enough to stand on its own. We also feel that the figure is too busy with all of its labels to be in the manuscript and that it should be a supplementary figure. Finally, we'd like to encourage readers to use the supplementary information as much as possible so that the entirety of our contribution can be used and appreciated by readers. Thus, we've opted to leave the figure as a supplementary figure and hope the reviewer is ok with this decision.

Reviewer #3

The purpose of this manuscript may be summarized by a statement given in the introduction: "This work introduces a new kind of metamaterial that enables such applications via user specified fluctuating Poisson's ratios." Toward that the authors provide a very rigorous modeling effort, combined with manufacture and experimental validation. The model and implementation are of top quality and very comprehensive. The verification with FEA is appreciated, specifically how it is used to determine the working limits of the model. The 2D and 3D experimental effort shows that this design functions well across length scales, and in multiple spatial dimension. Overall I believe there is a contribution to the literature here, the topic is of interest to readers from a variety of fields, and accordingly I recommend only minor revisions, and some specific comments are listed below.

We thank the reviewer for the careful review and generous assessment. We have revised the paper to address the discussed points below.

1) For some reason I think the authors can get a lot more traction by substituting the word fluctuating with something else. The definition I found of fluctuate is "rise and fall irregularly in number or amount." The irregular part is what gives me second thoughts. I concede this isn't necessarily how everyone will interpret the word, but this is just my opinion and does not need to be considered too heavily. Just think a different choice of word may be better, but I also cannot offer any appropriate suggestions. (The best I could think of was programmed mutation.)

We thank the reviewer for this helpful comment, and we fully agree that a better term should replace "fluctuating". We believe the term "alternating" better addresses the nature of our work, and we have made the appropriate replacements throughout. All instances have been highlighted green throughout the paper.

2) Around line 121, the authors mention a 60 minute period to keep things quasi-static. Generally speaking, the default in a FEA is a static analysis, which does not include any inertial terms, and therefore does not include any dynamics. But to point it out might imply that a dynamic analysis was used, and I think that should be mentioned in the manuscript.

The reviewer is correct. We apologize for the confusion. The manuscript has been updated and clarified where appropriate (pp. 6,11).

3) Around line 135 there is mention of hard stop 1 the only place where friction is expected. Is this friction an enemy to the desired behavior, or does it not matter and the choice of Teflon as the material not important?

This is a great point. We updated the manuscript to answer the question (pp. 7).

4) Around line 202, the authors mention that the properties of the constituent material are needed for the matlab tool. I would prefer if the properties required were explicitly mentioned, for example is it the thermal conductivity that is needed? Or just the elastic modulus? It is generally obvious, but never explicitly written.

We updated the manuscript to explicitly state what properties the tool requires as requested (pp.

5) There is mention of a 37 times faster calculation when compared to standard FEA. But I also wasn't able to determine if that was an elastic-plastic simulation or just an elastic simulation. Or a static or dynamic analysis. Those FEA details would help to make the comparison a bit more measurable to readers.

We updated the manuscript to provide more details about which simulations took what durations of time so that readers could get a better sense of the computational efficiency comparison between the MATLAB tool and the different FEA simulations for different designs (pp. 14).

6) Line 350 mentions an "in-situ pandoindenter." Since this is on the cutting edge I actually did a double take before realizing its a simple typo.

We confirmed this is a typo on our end, which has been appropriately corrected (pp. 19). We thank the reviewer for their careful review.

REVIEWERS' COMMENTS

Reviewer #1 (Remarks to the Author):

The manuscript could be accepted in the current form.

Reviewer #2 (Remarks to the Author):

The paper would benefit from having supplementary Figure 1 in the main body of the paper because it is essential to the information presented in the main body of the paper and is referenced by tables in the main body. Placing it as Sub-figure C to Figure 1 in the main body of the text seems like a good fit and the figure does not seem too busy or distracting from the other portions of the figure. This figure in the main body would better help the reader to understand the work that is being described. If there are editorial or length issues the reviewer is not aware of, this shouldn't be a show stopper. The paper is ready for publication.

Reviewer #3 (Remarks to the Author):

Thank you, all looks good.

RESPONSE TO REVIEWERS' COMMENTS

We are grateful to the reviewers for their insightful comments and for the time invested to help improve our paper. Reviewer comments are highlighted in dark blue and our responses follow. Page numbers are highlighted with the color used to highlight the corresponding revisions made on the corresponding page in the manuscript.

Reviewer #1 (Remarks to the Author):

The manuscript could be accepted in the current form.

We thank the reviewer for the time invested in making this paper good enough to be accepted.

Reviewer #2 (Remarks to the Author):

The paper would benefit from having supplementary Figure 1 in the main body of the paper because it is essential to the information presented in the main body of the paper and is referenced by tables in the main body. Placing it as Sub-figure C to Figure 1 in the main body of the text seems like a good fit and the figure does not seem too busy or distracting from the other portions of the figure. This figure in the main body would better help the reader to understand the work that is being described. If there are editorial or length issues the reviewer is not aware of, this shouldn't be a show stopper. The paper is ready for publication.

We learned that we can have more figures in the main text so we have added Supplementary Figure 1 as its own figure in the main text as suggested and updated all the figure references accordingly in all relevant documents. Thanks for your efforts to improve our paper. (pp. 7)

Reviewer #3 (Remarks to the Author):

Thank you, all looks good.

We appreciate the reviewer's help.